# Extended regimen of a levonorgestrel/ethinyl estradiol transdermal delivery system: Predicted serum hormone levels using a population pharmacokinetic model

**Frank Z. Stanczyk**[1]*, **David F. Archer**[2], **Lauren R. L. Lohmer**[3], **Jason Pirone**[3], **Michelle Previtera**[4], **Paul Korner**[4]

**1** Keck School of Medicine, University of Southern California, Los Angeles, Los Angeles, California, United States of America, **2** Eastern Virginia Medical School, Norfolk, Virginia, United States of America, **3** Nuventra, LLC, Durham, North Carolina, United States of America, **4** Agile Therapeutics, Inc., Princeton, New Jersey, United States of America

* fstanczyk@att.net

## Abstract

### Objective

This study employed population pharmacokinetic (popPK) models to predict levonorgestrel (LNG) and ethinyl estradiol (EE) exposure after dosing with the transdermal contraceptive TWIRLA® (LNG/EE TDS) as a 12-week extended regimen in a healthy female population.

### Methods

PopPK models were developed using data from a previously published phase 1, open-label, randomized clinical trial, ATI-CL14 (NCT01243580), in 36 healthy individuals. Models used cycle 2 data from 18 individuals who received the LNG/EE TDS, delivering LNG 120 µg/day and EE 30 µg/day, followed by a 1-week TDS-free period. Noncompartmental PK analyses were performed on simulated concentration–time profiles of 12 consecutive weeks of LNG/EE TDS use.

### Results

The simulated concentration–time profiles and PK parameters for the simulated extended regimen indicated that predicted LNG and EE exposures at week 12 were similar to week 3 (predicted geometric mean EE area under the concentration-time curve from time 0 to 168 h [$AUC_{0-168}$] on week 3 was 0.2% lower than week 12 and LNG $AUC_{0-168}$ on week 3 was 0.9% lower than week 12), suggesting both were at steady state by week 3. Therefore, no notable accumulation beyond that at week 3 is predicted for LNG and EE following a 12-week extended regimen. The results are supported by the accumulation ratios based on maximum concentration and the area under the curve being similar at weeks 3 and 12 for LNG and EE.

**Data Availability Statement:** All relevant data are within the manuscript and its Supporting Information files.

**Funding:** This study was funded by Agile Therapeutics, Inc., Princeton, NJ. Medical writing was provided by Chameleon Communications, New York, NY and was funded by Agile Therapeutics, Inc. Agile Therapeutics, Inc., provided a full review of the article and had a role in the design, execution, data collection and analysis, reporting, and funding of the study.

**Competing interests:** Competing interests FZS: Consultant: Agile Therapeutics. DFA: Consultant: Agile Therapeutics, Bayer Healthcare, Exeltis, Mithra, Lupin, ObsEva; Grants: Bayer Healthcare, Dare Biosciences, Estetra, Myovant, ObsEva; Honoraria: Exeltis; Patent: pending; Member: DSMB; Board member: Diczfalusy foundation; Stock ownership: InnovaGyn Inc, Agile Therapeutics LRLL: Consultant: Agile Therapeutics. JP: Consultant: Agile Therapeutics. MP: Employee: Agile Therapeutics; Stock ownership: Agile Therapeutics. PK: Employee: Agile Therapeutics; Consultant: Voltron Therapeutics; Independent board director: Voltron Therapeutics; Support for business-related travel: Agile Therapeutics; Stipend: Laidlaw Venture Partners LLC Investment and Operations Committee; Stock ownership: Agile Therapeutics, Voltron Therapeutics.

## Conclusion

These results indicate that a 12-week extended LNG/EE regimen would provide similar systemic hormonal exposure as that seen by week 3 in a standard 28-day regimen, without further hormonal accumulation. The data support the safe use of a non-daily, low-dose hormonal contraceptive in an extended regimen but should be confirmed in a clinical PK study.

## Introduction

Approximately half of women who use oral contraceptives exhibit poor adherence to their regimen, typically defined as missing at least 1 pill per month [1–4]. Poor adherence and discontinuation of contraception is responsible for roughly 20% of unintended pregnancies in the United States [5]. Selection of a contraceptive method associated with higher levels of treatment adherence is, therefore, likely to lower the risk of unintended pregnancy. Non-daily transdermal contraceptive formulations are associated with high levels of treatment adherence and patient satisfaction [6–8]. Additionally, a substantial number of women prefer to reduce the frequency of their menstrual periods, and a third of women would choose to never have a period [9]. Many women prefer to have less frequent menstrual bleeding [10]. Over the past 15 years, the US Food and Drug Administration (FDA) has approved several extended regimen oral contraceptives, including monophasic LNG/EE formulations and 1 quadriphasic LNG/EE combination [11–14].

TWIRLA® (AG200-15) (Agile Therapeutics, Inc., Princeton, NJ) is a low-dose contraceptive formulation of levonorgestrel (LNG) 2.6 mg and ethinyl estradiol (EE) 2.3 mg in a transdermal delivery system (TDS), approved by the US Food and Drug Administration (FDA) for women of reproductive potential with a body mass index (BMI) <30 kg/m$^2$. The LNG/EE TDS is approved for use as a 28-day-cycle contraceptive, with application and replacement every 7 days for 3 consecutive weeks, delivering LNG 120 µg/day and EE 30 µg/day, followed by a 7-day TDS-free period [15].The LNG/EE TDS avoids the need for a daily pill-taking regimen, as with oral contraceptives [15].

In this analysis, population pharmacokinetic (popPK) models were used to predict exposure to EE and LNG over 12 weeks (menstrual periods 4 times per year) of continuous LNG/EE TDS use in a healthy female population.

## Methods

### Study design

PK data employed in the present analysis were collected from a phase 1, open-label, single-center, randomized clinical trial, ATI-CL14 (NCT01243580), conducted over 3 28-day treatment cycles [16]. ATI-CL14 compared the LNG/EE TDS with a combination oral contraceptive containing EE 35 µg and norgestimate 250 µg, and enrolled 36 healthy, non-smoking women aged 18 to 45 years, with 24- to 35-day menstrual intervals. This study was approved by an Institutional Review Board, in accordance with the ethical principles the Declaration of Helsinki and International Conference on Harmonization Note for Guidance on Good Clinical Practice (CPMP/ICH/135/95). Written informed consent was obtained from subjects prior to the performance of any study-specific procedures. This study was conducted between 31 August 2009 and 07 December 2009. Participants were

normotensive and had a BMI between 18 and 32 kg/m$^2$. Exclusion criteria included individuals who were pregnant or breastfeeding; had a medical history contraindicating use of contraceptive steroids; or had dermal hypersensitivity, a recent abnormal cervical Pap test, or a positive hepatitis B/C or HIV antibody test [16]. During the first and third week of AG200-15 wear, samples for PK analysis were obtained immediately prior to TDS application and 6, 12, 24, 48, 72, 120, 144, and 168 h following TDS application. Following the third application, additional samples were collected at 174, 180, 192, 216, and 240 h. This is a total of 23 samples per subject in Cycle 2.

## Assays

Blood samples were kept frozen at approximately −80˚C before analysis. Analytes were isolated via extraction with organic solvent. The solvent was evaporated, and the remaining residue was derivatized. Derivatized analytes were extracted into hexane, which was evaporated, and the remaining residue was reconstituted with 300 μL of acetonitrile/water. The final extract was analyzed via high-performance liquid chromatography with tandem mass spectrometry detection. The internal standards used for LNG and EE assays were norgestrel-(ethyl-d5) and 17α-ethinylestradiol-2,4,16,16-d4, respectively. A previous bridging experiment demonstrated that either LNG or norgestrel reference material could be used with the method with consistent results, because the method does not chromatographically separate the enantiomers. This method is applicable to quantitation of LNG within a nominal range of 50 to 25,000 pg/mL and EE within a nominal range of 2 to 500 pg/mL and requires a 500-μL human plasma aliquot containing potassium oxalate/sodium fluoride. The ability to dilute samples originally above the upper limit of the calibration range and to analyze samples with insufficient volume for a full aliquot were validated. Validation showed no notable chromatographic peaks detected at the mass transitions and no notable expected retention times of the analytes or their internal standards that could interfere with quantitation. Validation also indicated that matrix suppression effects did not compromise the accuracy of the assay. Lower limits of quantification for the EE and LNG assays were 2 pg/mL and 50 pg/mL, respectively; the coefficients of variation were 5.17% to 15.3% and 2.28% to 15.8% for EE and LNG, respectively.

## PopPK models and simulations

**PopPK models.** Separate popPK models were developed for EE and LNG concentration–time data from ATI-CL14, based only on the cycle 2 data from individuals treated with the LNG/EE TDS in both cycles 1 and 2 (*N* = 18) in order to make predictions about EE and LNG PK following 12 consecutive weeks of TDS administration. No covariate analysis was performed for either analyte, given the small number of subjects. Phoenix NLME version 8.1 was used for model development and simulation. The decision to progress a model was based on inspection of standard goodness of fit plots, visual predictive checks (VPCs), and comparison of objective function values.

**EE popPK models.** A one-compartment model with zero-order infusion into the central compartment and first-order absorption from a dosing compartment adequately fit the data. Duration of the zero-order absorption into the central compartment was fixed to be 168 hours. Bioavailability was estimated based on a comparison of the patch exposure to the oral exposure with a known absolute bioavailability. Overall bioavailability (F) of EE was fixed (30%) due to identifiability issues with simultaneous estimation of F and relative bioavailability ($F_{rel}$); $F_{rel}$ was estimated for the separate absorption routes: $F_{rel} \times F$ and $(1-F_{rel}) \times F$ for the zero-order and first-order routes, respectively. Interindividual variability was included for volume, clearance, and the first-order absorption rate constant. Basic goodness-of-fit plots and a VPC

for the EE model are shown in S1A and S2A Figs, respectively. Goodness-of-fit plots suggest that the model adequately describes the observed data, with some underprediction of EE concentrations >0.1 ng/mL. Final model parameters were estimated with a relatively high degree of precision, with percent relative standard errors (RSEs) ranging from 7 to 14% (S1 Table).

**LNG popPK model.** A one-compartment model with zero-order absorption into the central compartment and zero-order infusion into a dosing compartment with first-order absorption adequately fit the data. Duration of the zero-order absorption into the central compartment was fixed to be 168 hours, and duration of the zero-order absorption into the dosing compartment was estimated by the model. Overall bioavailability (F) of LNG was fixed to 27% due to identifiability issues with simultaneous estimation of F and $F_{rel}$; $F_{rel}$ was estimated for the separate absorption routes: $F_{rel} \times F$ and $(1-F_{rel}) \times F$ for the first-order and zero-order routes, respectively. To describe variability in LNG PK between weeks, an effect on both volume and clearance at week 3 was included. Interindividual variability was included for volume, clearance, and $F_{rel}$ between the absorption routes. Basic goodness-of-fit plots and VPC for the LNG model are shown in S1B and S2B Figs, respectively. Goodness-of-fit plots suggest that the model adequately describes the observed data. The final parameter estimates for this model are summarized in S2 Table. As with the EE popPK model, final model parameters were estimated with fairly high precision, reflected in percent RSEs ranging from 10 to 31%, with the exception of week 3 effects on volume and clearance, which had percent RSEs of 57% and 47%, respectively.

## Extended regimen simulations

Simulations for the extended regimen of the LNG/EE TDS were conducted using the popPK models developed for LNG and EE. The proposed extended regimen consisted of 12 consecutive weeks (84 total days) of TDS use, delivering LNG 120 μg/day and EE 30 μg/day (2.6 mg LNG, 2.3 mg EE per TDS), followed by a 1-week (7-day) TDS-free period. LNG and EE concentrations were simulated for 1000 individuals every hour for weeks 1, 3, and 12. Interindividual variability was included in the simulations, but residual variability was not.

The popPK model and simulations did not take BMI or weight into consideration, although the BMI range of the individuals in ATI-CL14 was 18 to 32 kg/m$^2$, inclusive, consistent with the LNG/EE TDS indication.

## Noncompartmental analysis

Noncompartmental analysis was performed on simulated concentration–time profiles for weeks 1, 3, and 12, using Phoenix WinNonlin version 8.1. PK parameters were calculated for each week and for each simulated individual, as described in Table 1. R software version 3.4.0 and RStudio (version 1.1.163 or higher) environments were used for dataset preparation and generation of tables and figures.

## Results

### EE simulations

The popPK model developed for EE concentration–time data from study ATI-CL14 was used to simulate the EE concentration–time profile after 12 consecutive weeks of application [16].

**EE simulated concentration–time data.** Fig 1A shows the mean simulated concentration–time profiles for EE at weeks 1, 3, and 12, overlaid for the elapsed simulated time interval. Overlap of the mean concentrations of weeks 3 and 12 suggest that a steady state was achieved by week 3. S3 Table shows a summary of simulated concentrations at selected sampling

**Table 1. Pharmacokinetic parameters.**

| Parameter | Description |
|---|---|
| $C_{max}$ | Maximum concentration |
| $T_{max}$ | Time to maximum concentration |
| $AUC_{(0-168)}$ | Area under the plasma concentration–time curve from time 0 to 168 hours<br>• Calculated by a combination of linear and logarithmic trapezoidal methods (linear up/log down method) |
| AR AUC | Accumulation ratio based on area under the curve<br>• Calculated as $AUC_{(0-168)}$ (week 3 or week 12)/$AUC_{(0-168)}$ (week 1) |
| AR $C_{max}$ | Accumulation ratio based on $C_{max}$<br>• Calculated as $C_{max}$ (week 3 or week 12)/$C_{max}$ (week 1) |
| $C_{ss(0-168)}$ | Steady-state concentration from time 0 to 168 hours<br>• Calculated as $AUC_{(0-168)}$/168 |

timepoints (weeks 1, 3, 12). Mean EE concentrations ranged from 32.8 to 49.4 pg/mL during week 3 and from 33.0 to 49.5 pg/mL during week 12. The mean simulated EE concentration at weeks 3 and 12 peaked at 24 hours. Mean EE concentration at 168 hours during week 3 was 0.3% lower than that at 168 hours during week 12 (32.9 pg/mL vs 33.0 pg/mL).

**EE simulated noncompartmental PK parameters.** Noncompartmental PK parameters for EE at weeks 1, 3, and 12 are summarized in Table 2. Geometric mean estimates for maximum concentration ($C_{max}$) were 40.8, 47.9, and 48.0 pg/mL for weeks 1, 3, and 12, respectively, an increase from week 3 to 12 of 0.2%. Geometric mean accumulation ratio based on $C_{max}$ (AR $C_{max}$) was identical (1.17) at weeks 3 and 12 relative to week 1. The geometric mean values for steady-state concentration from 48 to 168 hours ($C_{SS(48-168)}$) were 33.5, 35.7, and 35.7 pg/mL for weeks 1, 3, and 12, respectively. Simulated $C_{max}$ and $C_{SS}$ values for EE suggested no notable accumulation of EE from week 3 through week 12. Box plots for EE area under the plasma concentration-time curve from time 0 to 168 hours ($AUC_{(0-168)}$) at weeks 1, 3, and 12 are shown in Fig 2A. Simulated EE $AUC_{(0-168)}$ geometric mean value for week 3 was 0.2% lower than that for week 12. Geometric mean AR based on $AUC_{(0-168)}$ (AR AUC) was 1.17 and 1.18 at weeks 3 and 12, respectively, suggesting no notable accumulation at week 12 relative to week 3.

## LNG simulations

As with the EE simulations, the popPK model developed using LNG concentration–time data from ATI-CL14 was employed to simulate LNG concentration–time profiles after 12 weeks of application [16].

**LNG simulated concentration–time data.** Mean simulated concentration–time profiles for LNG at weeks 1, 3, and 12, overlaid for the elapsed simulated time interval, are shown in Fig 1B. Mean concentrations at weeks 3 and 12 overlap, suggesting that steady-state concentration for LNG is predicted to be achieved by week 3. Simulated concentrations by week at selected sampling timepoints are summarized in S4 Table. Mean LNG concentrations ranged from 1410 to 2430 pg/mL during week 3 and from 1700 to 2470 pg/mL during week 12. Mean simulated LNG concentration at 168 hours during week 3 was 1.2% lower than that during week 12.

**LNG simulated noncompartmental PK parameters.** The noncompartmental PK parameters for LNG at weeks 1, 3, and 12 are summarized in Table 3. Geometric mean estimates for $C_{max}$ were 1420, 2310, and 2330 pg/mL for weeks 1, 3, and 12, respectively, representing a 0.9% increase from week 3 to week 12. AR $C_{max}$ relative to week 1 was 1.63 and 1.64 for weeks 3 and 12, respectively, and geometric mean values for $C_{SS(48-168)}$ were 1250, 1840, and 1860

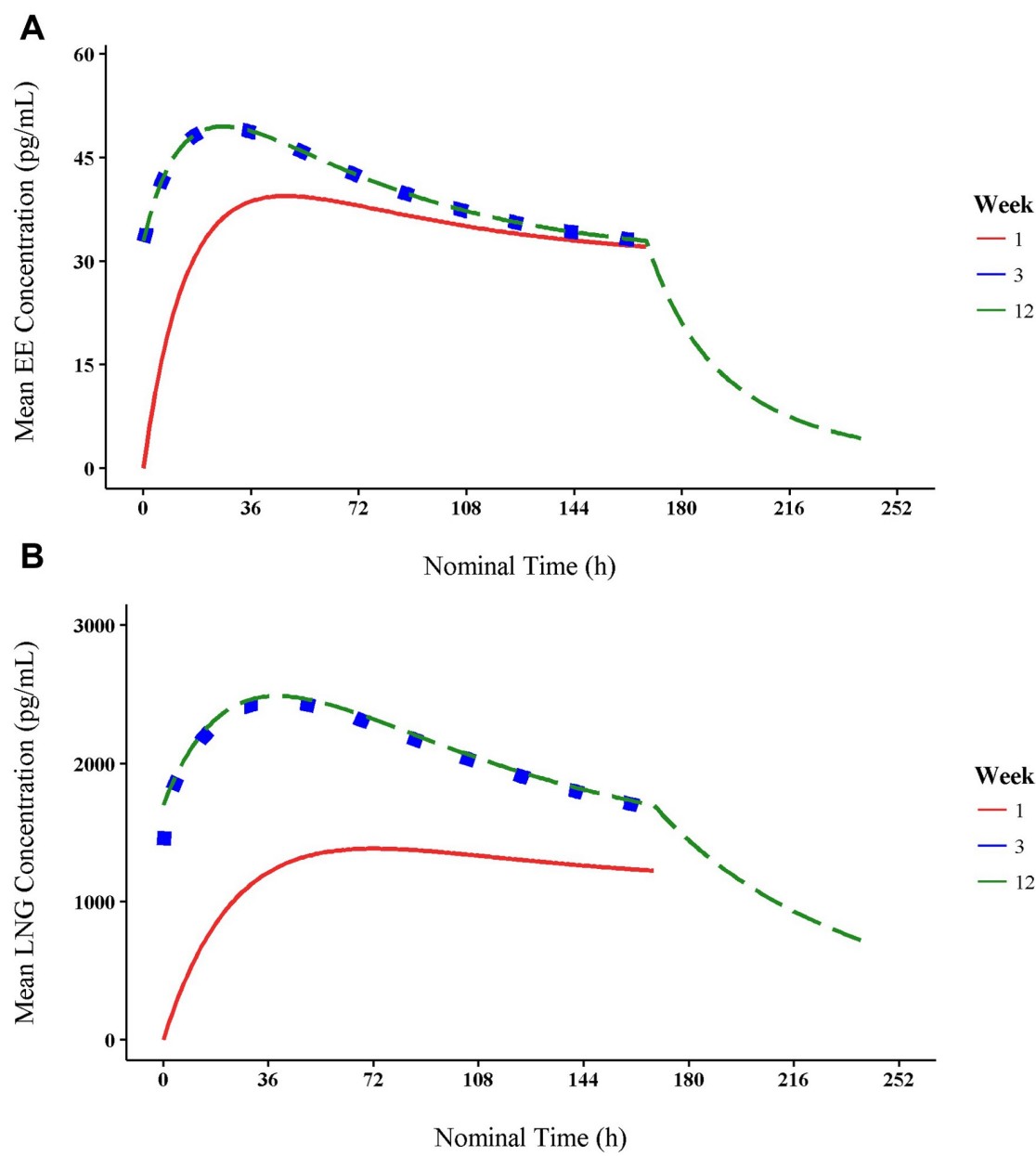

**Fig 1. Extended regimen mean simulated concentration-time profiles for EE and LNG at 1, 3 and 12 weeks.** Mean concentration–time profile of (A) EE and (B) LNG for extended regimen for weeks 1, 3, and 12, overlaid for the elapsed simulated time interval. EE, ethinyl estradiol; LNG, levonorgestrel.

pg/mL for weeks 1, 3, and 12 respectively. Simulated $C_{max}$ and $C_{SS}$ values for LNG indicate predicted exposure at week 12 is similar to that at week 3.

Fig 2B shows box plots of simulated LNG $AUC_{(0-168)}$ for an extended regimen by week. Simulated LNG $AUC_{(0-168)}$ geometric mean value for week 3 was 0.9% lower than that for week 12. Geometric mean AR AUC was 1.68 and 1.70 for weeks 3 and 12, respectively, suggesting no notable accumulation at week 12 relative to week 3.

**Table 2. Summary of EE pharmacokinetic parameters for extended-cycle regimen by week.**

| Week | Statistic | $C_{max}$ (pg/mL) | $T_{max}$ (h) | $AUC_{(0-168)}$ (h·ng/mL) | $C_{ss(0-168)}$ (pg/mL) | $C_{ss(48-168)}$ (pg/mL) | AR $C_{max}$ | AR AUC |
|---|---|---|---|---|---|---|---|---|
| 1 | N | 1000 | 1000 | 1000 | 1000 | 1000 | 0 | 0 |
| | Mean | 42.6 | NC | 5.72 | 34.0 | 35.4 | NC | NC |
| | SD | 12.6 | NC | 1.73 | 10.3 | 12.0 | NC | NC |
| | CV% | 29.5 | NC | 30.3 | 30.3 | 33.8 | NC | NC |
| | Geometric mean | 40.8 | NC | 5.47 | 32.6 | 33.5 | NC | NC |
| | Geometric CV% | 29.7 | NC | 30.3 | 30.3 | 34.2 | NC | NC |
| 3 | N | 1000 | 1000 | 1000 | 1000 | 1000 | 1000 | 1000 |
| | Mean | 50.3 | NC | 6.81 | 40.6 | 38.2 | 1.18 | 1.18 |
| | SD | 16.2 | NC | 2.44 | 14.5 | 14.6 | 0.0796 | 0.129 |
| | CV% | 32.2 | NC | 35.7 | 35.7 | 38.3 | 6.77 | 10.9 |
| | Geometric mean | 47.9 | NC | 6.42 | 38.2 | 35.7 | 1.17 | 1.17 |
| | Geometric CV% | 32.3 | NC | 35.4 | 35.4 | 38.1 | 6.48 | 9.96 |
| 12 | N | 1000 | 1000 | 1000 | 1000 | 1000 | 1000 | 1000 |
| | Mean | 50.4 | NC | 6.82 | 40.6 | 38.2 | 1.18 | 1.18 |
| | SD | 16.2 | NC | 2.45 | 14.6 | 14.7 | 0.0816 | 0.132 |
| | CV% | 32.2 | NC | 35.8 | 35.8 | 38.4 | 6.93 | 11.2 |
| | Geometric mean | 48.0 | NC | 6.43 | 38.3 | 35.7 | 1.17 | 1.18 |
| | Geometric CV% | 32.3 | NC | 35.4 | 35.4 | 38.2 | 6.61 | 10.2 |

AUC, area under the concentration curve; AR, accumulation ratio; $C_{max}$, maximum concentration; $C_{ss}$, steady-state concentration; CV%, arithmetic percent coefficient of variation; geometric CV%, geometric percent coefficient of variation; EE, ethinyl estradiol; mean, arithmetic mean; N, sample size; NC, not calculated; SD, standard deviation; $T_{max}$, time to maximum concentration.

## Discussion

The ideal characteristics of a contraception formulation are safety, efficacy, and utility. Utility comprises several qualities, including ease of use, convenience, a propensity to promote treatment adherence to reduce the risk of unintended pregnancy, and, for many women, reduction in number, intensity, and length of menstrual periods [17]. The LNG/EE TDS, applied once weekly, is well tolerated, with a safety profile similar to that observed in similar hormonal contraceptives, a high level of efficacy in preventing pregnancy in women with a BMI <30 km/mg², and a high rate of treatment adherence [18, 19]. A LNG/EE TDS extended regimen is likely to be of interest to those who prefer fewer menstrual cycles during the year using a non-daily contraceptive. To assess whether there is potential for hormonal accumulation, the present analysis was performed to predict LNG and EE concentrations following simulated administration of the LNG/EE TDS as an extended, 12-week regimen (84 total days). The popPK models for LNG and EE simulation utilized cycle 2 data from a cohort of 18 individuals applying the LNG/EE TDS for 2 consecutive cycles in a previously conducted clinical pharmacology study. These models were used to predict LNG and EE PK parameters over the extended regimen.

Data from only cycle 2 of subjects that received consecutive cycles of TDS application from Study ATI-CL14 were selected for popPK model development to represent the most conservative estimate of exposure in these simulations, as multiple dose studies demonstrated within-cycle and between-cycle increases in LNG and EE exposure [16, 20]. Upon removal of the TDS, serum levels of LNG and EE reach low and non-measurable levels within 3 days, respectively [20]. The between-cycle accumulation is not solely based on the $t_{1/2}$ of EE/LNG (as 168 h would be a full washout for either) but is possibly related to physiological changes (i.e., changes

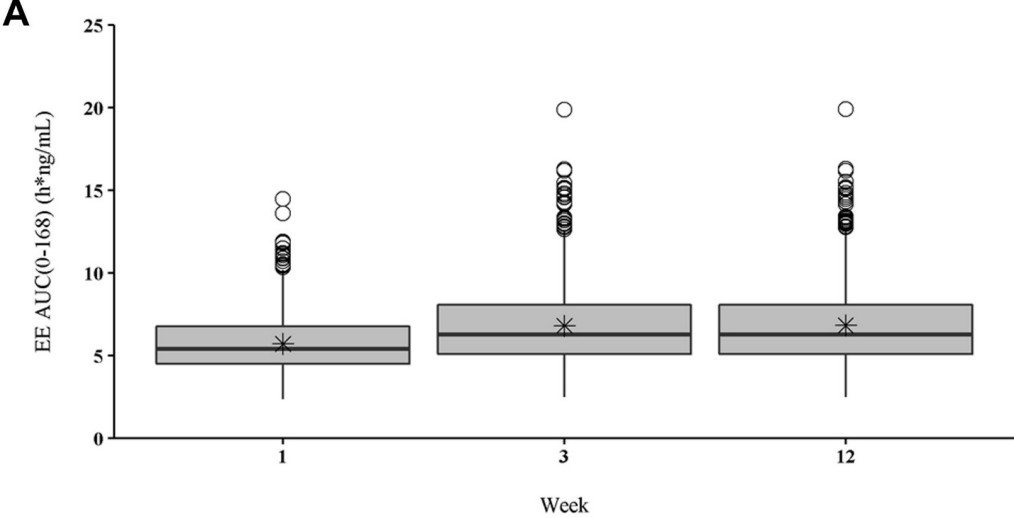

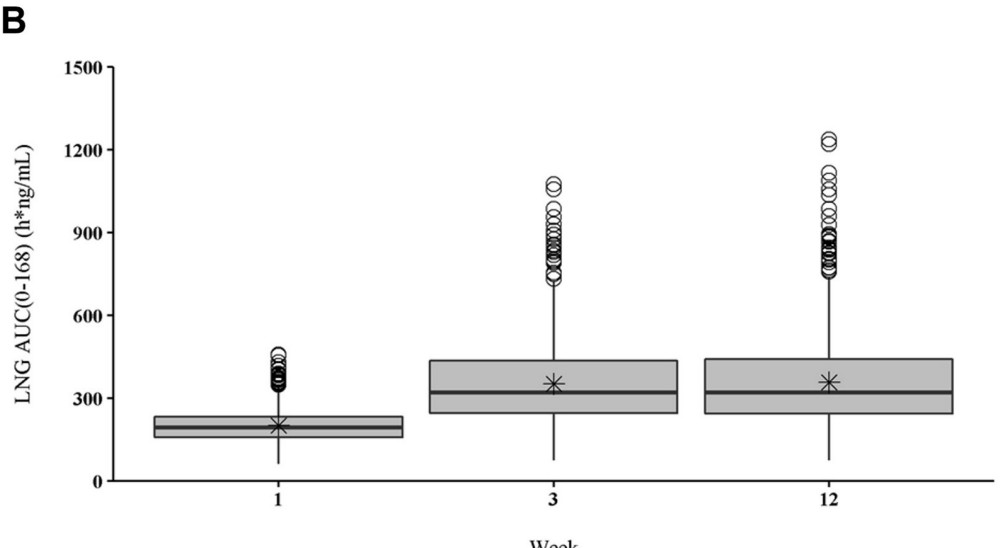

**Fig 2. Extended regimen plasma concentration-time curves for EE and LNG from 0 to 168 hours.** Box plots of (A) EE $AUC_{(0-168)}$ and (B) LNG $AUC_{(0-168)}$ for extended regimen at weeks 1, 3, and 12. Solid line = median; box edges = 25% and 75% quartiles (IQR); whiskers = lowest/highest values within 1.5× IQR of the lower/upper quartiles; circles = observations beyond whiskers; * = mean. AUC(0–168), area under the concentration curve from 0 to 168 hours; EE, ethinyl estradiol; LNG, levonorgestrel.

in levels of sex hormone-binding globulin, which LNG binds and EE induces). Evaluation of sex hormone-binding globulin in the context of an extended regimen TDS will help elucidate the impact of sex hormone-binding globulin on LNG attainment of steady state. In a three-cycle study, steady-state was reached during cycle 2.

Weight or body mass index are thought to be some of the factors that may lead to differences in LNG and EE PK [21]. Weight and body mass index could not be evaluated in this analysis due to the small number of subjects included (N = 18).

Results of the simulated concentration–time profiles suggest that predicted exposure at week 12 is similar to that at week 3 for both LNG and EE. popPK analysis predicted a small

**Table 3. Summary of LNG pharmacokinetic parameters for extended-cycle regimen by week.**

| Week | Statistic | $C_{max}$ (pg/mL) | $T_{max}$ (h) | $AUC_{(0-168)}$ (h·ng/mL) | $C_{ss(0-168)}$ (pg/mL) | $C_{ss(48-168)}$ (pg/mL) | AR $C_{max}$ | AR AUC |
|---|---|---|---|---|---|---|---|---|
| 1 | N | 1000 | 1000 | 1000 | 1000 | 1000 | 0 | 0 |
| | Mean | 1490 | NC | 201 | 1200 | 1320 | NC | NC |
| | SD | 461 | NC | 62.6 | 373 | 439 | NC | NC |
| | CV% | 31.0 | NC | 31.1 | 31.1 | 33.3 | NC | NC |
| | Geometric mean | 1420 | NC | 192 | 1140 | 1250 | NC | NC |
| | Geometric CV% | 31.1 | NC | 31.6 | 31.6 | 34.6 | NC | NC |
| 3 | N | 1000 | 1000 | 1000 | 1000 | 1000 | 1000 | 1000 |
| | Mean | 2470 | NC | 352 | 2100 | 2030 | 1.64 | 1.72 |
| | SD | 913 | NC | 150 | 894 | 917 | 0.230 | 0.395 |
| | CV% | 37.0 | NC | 42.7 | 42.7 | 45.2 | 14.0 | 22.9 |
| | Geometric mean | 2310 | NC | 323 | 1920 | 1840 | 1.63 | 1.68 |
| | Geometric CV% | 37.2 | NC | 43.6 | 43.6 | 46.9 | 13.8 | 21.4 |
| 12 | N | 1000 | 1000 | 1000 | 1000 | 1000 | 1000 | 1000 |
| | Mean | 2500 | NC | 358 | 2130 | 2060 | 1.66 | 1.75 |
| | SD | 992 | NC | 164 | 974 | 987 | 0.292 | 0.482 |
| | CV% | 39.6 | NC | 45.7 | 45.7 | 47.9 | 17.6 | 27.5 |
| | Geometric mean | 2330 | NC | 326 | 1940 | 1860 | 1.64 | 1.70 |
| | Geometric CV% | 38.8 | NC | 45.4 | 45.4 | 48.5 | 16.4 | 24.1 |

AUC, area under the concentration curve; AR, accumulation ratio; $C_{max}$, maximum concentration; $C_{ss}$, steady-state concentration; CV%, arithmetic percent coefficient of variation; geometric CV%, geometric percent coefficient of variation; LNG, levonorgestrel; mean, arithmetic mean; NC, not calculated; SD, standard deviation; $T_{max}$, time to maximum concentration.

increase in EE concentration from week 1 to week 3 (consistent with previously conducted clinical pharmacology study results), followed by steady-state achievement, with no accumulation of EE past week 3. This is consistent with previously published PK results where a small increase in EE concentration from week 1 to week 3 was observed [16]. The ARs based on $C_{max}$ and AUC for weeks 3 and 12 for both LNG and EE indicated that steady-state concentrations were achieved by week 3, and that there was no notable accumulation of LNG or EE during administration of the LNG/EE TDS as a 12-week extended regimen. The small, predicted increase in LNG and EE is unlikely to be clinically relevant given the observed variability in LNG and EE PK within and between cycles [20].

Lack of additional accumulation beyond that observed at week 3 is consistent with the observed PK for a once-daily combined oral contraceptive consisting of EE 30 µg/day and LNG 150 µg/day LNG (Seasonique®, Duramed Pharmaceuticals, Inc., Pomona, NY) after 84 consecutive days of dosing [22]. A separate combined oral contraceptive (LNG/EE and EE), employing an ascending-dose extended regimen (Quartette®, Teva Pharmaceuticals USA, Inc., North Wales, PA) and for which a popPK model predicted a stepwise increase in systemic exposure to EE during the first 84 days of a cycle after each increased EE dose change, has been shown to provide effective pregnancy prevention [14, 23]. Based on the simulations described herein, the LNG/EE TDS 12-week extended regimen is predicted to deliver overall lower exposures at steady-state EE and LNG concentrations compared with Quartette, while maintaining the same potential to provide effective pregnancy prevention as the approved 28-day regimen.

The limitations of this study are inherent in the simulation study design and the use of data from a single treatment cycle. Consistency of these outcomes with previous contraceptive

studies using similar formulations supports the predictive value of this methodology, although these results will require confirmation in a clinical PK study.

## Supporting information

**S1 Fig. Extended regimen popPK model goodness-of-fit plots for EE and LNG.** A. EE popPK model goodness-of-fit plots. B. LNG popPK model goodness-of-fit plots. EE, ethinyl estradiol; LNG, levonorgestrel; popPK, population pharmacokinetics.
(TIF)

**S2 Fig. Extended regimen visual predictive check for EE and LNG.** A. Visual Predictive Check for EE. B. Visual Predictive Check for LNG. EE, ethinyl estradiol; LNG, levonorgestrel.
(TIF)

**S1 Table. EE population pharmacokinetic final parameter estimates.**
(TIF)

**S2 Table. LNG population pharmacokinetic final parameter estimates.**
(TIF)

**S3 Table. Summary of EE concentration–time data for extended regimen by week.**
(TIF)

**S4 Table. Summary of LNG concentration–time data for extended regimen by week.**
(TIF)

## Acknowledgments

Medical writing support was provided by Mandakini Singh, PhD, of Chameleon Communications, New York, NY, funded by Agile Therapeutics, Inc., Princeton, NJ. This manuscript was prepared according to the International Society for Medical Publication Professionals' "Good Publication Practice for Communicating Company-Sponsored Medical Research: GPP3."

## Author Contributions

**Conceptualization:** Frank Z. Stanczyk.

**Data curation:** Lauren R. L. Lohmer, Jason Pirone, Michelle Previtera, Paul Korner.

**Formal analysis:** Lauren R. L. Lohmer, Jason Pirone, Michelle Previtera, Paul Korner.

**Writing – original draft:** Frank Z. Stanczyk, David F. Archer, Lauren R. L. Lohmer, Jason Pirone, Michelle Previtera, Paul Korner.

**Writing – review & editing:** Frank Z. Stanczyk, David F. Archer, Lauren R. L. Lohmer, Jason Pirone, Michelle Previtera, Paul Korner.

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
