## [Decision Letter · Decision Letter 0]

26 Sep 2022

PONE-D-22-16688Extended regimen of a levonorgestrel/ethinyl estradiol transdermal delivery system: predicted serum hormone levels using a population pharmacokinetic modelPLOS ONE

Dear Dr. Stanczyk,

Thank you for submitting your manuscript to PLOS ONE. After careful consideration, we feel that it has merit but does not fully meet PLOS ONE’s publication criteria as it currently stands. Therefore, we invite you to submit a revised version of the manuscript that addresses the points raised during the review process.

The manuscript has been evaluated by three reviewers, and their comments are available below.

The reviewers have raised a number of concerns that need attention. They request additional information on the potential to integrate other covariates into the model and study dynamics of LNG/EE TDS, and on the source of the data used in the analysis. They also suggest an expansion on the possible impact of sex hormone-binding globulin on LNG concentrations or incorporating this into the modeling and analysis. 

Could you please revise the manuscript to carefully address the concerns raised?

We look forward to receiving your revised manuscript.

Kind regards,

Alice Coles-Aldridge

Editorial Office

PLOS ONE

Journal Requirements:

3. Please provide additional details regarding participant consent. In the ethics statement in the Methods and online submission information, please ensure that you have specified what type you obtained (for instance, written or verbal, and if verbal, how it was documented and witnessed). If your study included minors, state whether you obtained consent from parents or guardians. If the need for consent was waived by the ethics committee, please include this information

Reviewers' comments:

Reviewer's Responses to Questions

**Comments to the Author**

1. Is the manuscript technically sound, and do the data support the conclusions?

Reviewer #1: Yes

Reviewer #2: Yes

Reviewer #3: Yes

2. Has the statistical analysis been performed appropriately and rigorously? 

Reviewer #1: Yes

Reviewer #2: Yes

Reviewer #3: No

3. Have the authors made all data underlying the findings in their manuscript fully available?

Reviewer #1: Yes

Reviewer #2: Yes

Reviewer #3: No

4. Is the manuscript presented in an intelligible fashion and written in standard English?

Reviewer #1: Yes

Reviewer #2: Yes

Reviewer #3: Yes

5. Review Comments to the Author

Reviewer #1: A population pharmacokinetic (popPK) model was used to predict levonorgestrel (LNG) and ethinyl estradiol (EE) exposure after dosing with the TWIRLA contraceptive from a phase 1 trial. The area under the curve for LNG, and for EE, was similar at 3 and 12 weeks. The mathematical/statistical methods were well described and thorough.

Minor revision:

Indicate the date range subjects participated in the study.

Reviewer #2: Overall structure:

The study topic is of relevance and general interest to the readers of the journal. Overall, the paper is well written. The authors focus on population pharmacokinetic (PopPK) models to predict LNG and EE exposure after dosing with the TDS over 12 weeks period of continuous use in a healthy female population. The various analysis and specifically the PopPK models applied to respond to the main objective of the study makes the dataset quite useful. However, it would have been interesting to integrate other covariates like age, race, etc. in the model and study any glaring dynamics of the LNG/EE TDS, thus leaving room for interesting inquiries to be able to recommend the paper.

Introduction:

- The introduction section has ably captured the problem statement and carefully linked the interest in PopPK modeling to understand and predict the exposure to EE and LNG.

Methods:

- The methodology is clearly put out and is reproducible for similar studies that could probably be interested in understanding other covariate integrations and dynamics not listed.

Discussion:

- The discussion section is well thought out and presented links to the objective of the study and also aligning its findings to previous similar studies

Reviewer #3: Data availability is limited to data points in diagnostic plots. Is this sufficient?

Some additional information could be included regarding the source of data used in the analysis - only cycle 2 data was selected, was there a reason behind this? Is the model in agreement with cycle 1 or cycle 3 results?

The reference to the possible impact of sex hormone-binding globulin on LNG concentrations is interesting and could be expanded on or incorporated into the modeling and analysis to explore further. Maybe worth folding into this work or suggesting for future analyses.

Was BMI and/or body weight explored to determine there weren't enough subjects with n=18? If so, would be good to share the results that support the decision.

6. PLOS authors have the option to publish the peer review history of their article (what does this mean?). If published, this will include your full peer review and any attached files.

Reviewer #1: No

Reviewer #2: No

Reviewer #3: No

---

## [Author Response · Author response to Decision Letter 0]

1 Nov 2022

Reviewer #1

A population pharmacokinetic (popPK) model was used to predict levonorgestrel (LNG) and ethinyl estradiol (EE) exposure after dosing with the TWIRLA contraceptive from a phase 1 trial. The area under the curve for LNG, and for EE, was similar at 3 and 12 weeks. The mathematical/statistical methods were well described and thorough.

Minor revision:

Indicate the date range subjects participated in the study.

The dates of study execution have been added to the Methods. 

Reviewer #2

Overall structure:

The study topic is of relevance and general interest to the readers of the journal. Overall, the paper is well written. The authors focus on population pharmacokinetic (PopPK) models to predict LNG and EE exposure after dosing with the TDS over 12 weeks period of continuous use in a healthy female population. The various analysis and specifically the PopPK models applied to respond to the main objective of the study makes the dataset quite useful. However, it would have been interesting to integrate other covariates like age, race, etc. in the model and study any glaring dynamics of the LNG/EE TDS, thus leaving room for interesting inquiries to be able to recommend the paper.

We agree that evaluation of covariates on the PK of LNG and EE is an interesting and important line of inquiry. Given the small number subjects included, it was not feasible in the present analysis. 

Introduction:

- The introduction section has ably captured the problem statement and carefully linked the interest in PopPK modeling to understand and predict the exposure to EE and LNG.

Methods:

- The methodology is clearly put out and is reproducible for similar studies that could probably be interested in understanding other covariate integrations and dynamics not listed.

Discussion:

- The discussion section is well thought out and presented links to the objective of the study and also aligning its findings to previous similar studies

Reviewer #3

Data availability is limited to data points in diagnostic plots. Is this sufficient?

Data available upon request. 

Some additional information could be included regarding the source of data used in the analysis - only cycle 2 data was selected, was there a reason behind this? Is the model in agreement with cycle 1 or cycle 3 results?

Only cycle 2 data was selected for this analysis to enable the most conservative estimate of LNG and EE exposure after the extended regimen TDS application. As in the Discussion, there are known within-cycle and between-cycle increases in LNG and EE exposure that are not fully attributable to the terminal elimination half life of these hormones. The subjects treated in cycle 2 with the TDS were administered 2 sequential cycles of TDS, which has been clarified in the Methods.

The reference to the possible impact of sex hormone-binding globulin on LNG concentrations is interesting and could be expanded on or incorporated into the modeling and analysis to explore further. Maybe worth folding into this work or suggesting for future analyses.

We agree with the effect of sex hormone-binding globulin on LNG concentration is an important aspect of LNG pharmacokinetics. Sex hormone-binding globulin was not collected in this study and could not be incorporated into the model. An additional comment on evaluation of sex hormone-binding globulin on LNG pharmacokinetics in the context of extended regimen TDS administration has been added to the discussion.

Was BMI and/or body weight explored to determine there weren't enough subjects with n=18? If so, would be good to share the results that support the decision.

We agree that evaluation of covariates on the PK of LNG and EE is an interesting and important line of inquiry. Given the small number subjects included, it was not feasible in the present analysis. 

Additional requirements

Please ensure that your manuscript meets PLOS ONE's style requirements, including those for file naming. The PLOS ONE style templates can be found at https://journals.plos.org/plosone/s/file?id=wjVg/PLOSOne_formatting_sample_main_body.pdf and https://journals.plos.org/plosone/s/file?id=ba62/PLOSOne_formatting_sample_title_authors_affiliations.pdf

The manuscript meets PLOS ONE’S style requirements. 

PLOS requires an ORCID iD for the corresponding author in Editorial Manager on papers submitted after December 6th, 2016. Please ensure that you have an ORCID iD and that it is validated in Editorial Manager. To do this, go to ‘Update my Information’ (in the upper left-hand corner of the main menu), and click on the Fetch/Validate link next to the ORCID field. This will take you to the ORCID site and allow you to create a new iD or authenticate a pre-existing iD in Editorial Manager. Please see the following video for instructions on linking an ORCID iD to your Editorial Manager account: https://www.youtube.com/watch?v=_xcclfuvtxQ

ORCID has been linked in PLOS ONE submission manager.

Please provide additional details regarding participant consent in the ethics statement in the Methods and online submission information, please ensure that you have specified what type you obtained (for instance, written or verbal, and if verbal, how it was documented and witnessed). If your study included minors, state whether you obtained consent from parents or guardians. If the need for consent was waived by the ethics committee, please include this information.

Information regarding ethical conduct of the study and written informed consent has been added to the Methods. 

We note that the grant information you provided in the ‘Funding Information’ and ‘Financial Disclosure’ sections do not match. When you resubmit, please ensure that you provide the correct grant numbers for the awards you received for your study in the ‘Funding Information’ section.

This will be reconciled upon resubmission.

---

## [Decision Letter · Decision Letter 1]

12 Dec 2022

Extended regimen of a levonorgestrel/ethinyl estradiol transdermal delivery system: predicted serum hormone levels using a population pharmacokinetic model

PONE-D-22-16688R1

Dear Dr. Stanczyk,

We’re pleased to inform you that your manuscript has been judged scientifically suitable for publication and will be formally accepted for publication once it meets all outstanding technical requirements.

Kind regards,

George Vousden

Staff Editor

PLOS ONE

Additional Editor Comments (optional):

Reviewers' comments:

Reviewer's Responses to Questions

**Comments to the Author**

1. If the authors have adequately addressed your comments raised in a previous round of review and you feel that this manuscript is now acceptable for publication, you may indicate that here to bypass the “Comments to the Author” section, enter your conflict of interest statement in the “Confidential to Editor” section, and submit your "Accept" recommendation.

Reviewer #1: All comments have been addressed

Reviewer #2: All comments have been addressed

Reviewer #3: All comments have been addressed

2. Is the manuscript technically sound, and do the data support the conclusions?

Reviewer #1: (No Response)

Reviewer #2: Yes

Reviewer #3: Yes

3. Has the statistical analysis been performed appropriately and rigorously? 

Reviewer #1: (No Response)

Reviewer #2: Yes

Reviewer #3: Yes

4. Have the authors made all data underlying the findings in their manuscript fully available?

Reviewer #1: (No Response)

Reviewer #2: Yes

Reviewer #3: Yes

5. Is the manuscript presented in an intelligible fashion and written in standard English?

Reviewer #1: (No Response)

Reviewer #2: Yes

Reviewer #3: Yes

6. Review Comments to the Author

Reviewer #1: (No Response)

Reviewer #2: (No Response)

Reviewer #3: (No Response)

7. PLOS authors have the option to publish the peer review history of their article (what does this mean?). If published, this will include your full peer review and any attached files.

Reviewer #1: No

Reviewer #2: No

Reviewer #3: **Yes: **Joshuaine Grant

---

## [Editor Report · Acceptance letter]

15 Dec 2022

PONE-D-22-16688R1 

Extended regimen of a levonorgestrel/ethinyl estradiol transdermal delivery system: predicted serum hormone levels using a population pharmacokinetic model 

Dear Dr. Stanczyk:

I'm pleased to inform you that your manuscript has been deemed suitable for publication in PLOS ONE. Congratulations! Your manuscript is now with our production department. 

Kind regards, 

on behalf of

Dr. George Vousden 

Staff Editor

PLOS ONE